# Tetraethylenepentamine-Coated β Cyclodextrin Nanoparticles for Dual DNA and siRNA Delivery

**DOI:** 10.3390/pharmaceutics14050921

**Published:** 2022-04-23

**Authors:** Chi-Hsien Liu, Pei-Yin Shih, Cheng-Han Lin, Yi-Jun Chen, Wei-Chi Wu, Chun-Chao Wang

**Affiliations:** 1Department of Chemical and Materials Engineering, Chang Gung University, 259, Wen-Hwa First Road, Kwei-Shan, Taoyuan 33302, Taiwan; fengric159@yahoo.com.tw (C.-H.L.); a0958860407@gmail.com (Y.-J.C.); 2Research Center for Chinese Herbal Medicine and Research Center for Food and Cosmetic Safety, College of Human Ecology, Chang Gung University of Science and Technology, 261, Wen-Hwa First Road, Taoyuan 33302, Taiwan; 3Department of Chemical Engineering, Ming Chi University of Technology, 84, Gung-Juan Road, New Taipei City 24301, Taiwan; 4Department of Ophthalmology, Chang Gung Memorial Hospital, Linkou, 5, Fu-Hsing Street, Taoyuan 33305, Taiwan; weichi666@gmail.com; 5Graduate Institute of Biochemical and Biomedical Engineering, Chang Gung University, Taoyuan 33302, Taiwan; sunshine850215@gmail.com; 6College of Medicine, Chang Gung University, Taoyuan 33302, Taiwan; 7Institute of Molecular Medicine & Department of Medical Science, National Tsing Hua University, 101, Kuang-Fu Road, Hsinchu 30013, Taiwan

**Keywords:** β-cyclodextrin, siRNA, knockdown, transfection, nanoparticle

## Abstract

Nucleic acid reagents, including plasmid-encoded genes and small interfering RNA (siRNA), are promising tools for validating gene function and for the development of therapeutic agents. Native β-cyclodextrins (BCDs) have limited efficiency in gene delivery due to their instable complexes with nucleic acid. We hypothesized that cationic BCD nanoparticles could be an efficient carrier for both DNA and siRNA. Tetraethylenepentamine-coated β-cyclodextrin (TEPA-BCD) nanoparticles were synthesized, characterized, and evaluated for targeted cell delivery of plasmid DNA and siRNA. The cationic TEPA coating provided ideal zeta potential and effective nucleic acid binding ability. When transfecting plasmid encoding green fluorescent protein (GFP) by TEPA-BCD, excellent GFP expression could be achieved in multiple cell lines. In addition, siRNA transfected by TEPA-BCD suppressed target GFP gene expression. We showed that TEPA-BCD internalization was mediated by energy-dependent endocytosis via both clathrin-dependent and caveolin-dependent endocytic pathways. TEPA-BCD nanoparticles provide an effective means of nucleic acid delivery and can act as potential carriers in future pharmaceutical application.

## 1. Introduction

Gene expression could be modulated through the use of exogenous nucleic acids. The power of nucleic-acid-based drugs, including DNA molecules and small interfering RNAs (siRNAs), lies in their abilities to specifically enhance or silence genes of interest. The siRNA-based drugs were recently approved by US Federal Drug Administration [1], marking a milestone for new types of therapeutic strategies [2]. The small size and facile synthesis of siRNA give it some advantages over DNA and protein drugs. However, the “naked” siRNA is not stable and can be degraded by nucleases. siRNA is unable to diffuse through cell membranes due to its anionic charge. The major barriers to siRNA clinical application include the lack of efficient nanocarriers, rapid clearance from systemic circulation, targeted delivery, limited cellular uptake, and inability to traffic efficiently to the cytoplasm of cells [3,4]. There is a need for new delivery tools that can endow siRNAs against degradation, high payload, and good silencing efficiency [5,6].

Nonviral gene delivery vectors have been shown to have several advantages, such as ease of synthesis, mass production, versatile modification, and low immunogenicity [7]. There are many approaches being taken to the development of siRNA nanocarriers [8,9,10]. Cyclodextrin polymers offer tunable functional properties, including particle size, zeta potential, and targeted ligand conjugation [11,12]. The cone-shaped β-cyclodextrin (BCD) nanoparticle, composed of seven α-(1 → 4)-linked D-glucopyranoside subunits, has been known as transfection enhancer [13]. The hydroxyl groups on BCD can also be functionally modified to obtain amphiphilic, anionic, and cationic derivatives [14]. These BCD can increase cellular membrane permeability of both DNA and siRNA, and are capable of delivering them efficiently. There has been growing interest in the development of BCD as a delivery vehicle for nucleic acids. For example, several researchers have developed cyclodextrin-containing polymers as siRNA vehicles to treat metastatic cancers [15,16,17]. However, studies related to tetraethylenepentamine (TEPA)-modified polymers as carriers for pDNA and siRNA are rare, according to our literature survey. The safety of BCD as drug carriers for therapeutic delivery is well established [18].

Cationic polysaccharides can bind with siRNAs to form complexes, thereby protecting siRNA from degradation and neutralizing their negative charges [19]. Nanoparticles formed with cationic BCD and siRNA demonstrated efficacy not only in knocking down genes of interest in mice [20], but were able to inhibit tumor growth in a mouse model of metastatic Ewing’s sarcoma [21]. The gold standard nonviral vector, polyethylenimine (PEI), is an efficient nucleic acid delivery polycation. However, its transfecting activity is accompanied by toxicity. Thus, PEI has limited use in clinical practice. The siRNA compaction and transfection activity could be increased with the carriers containing variable cationic polyamines, of which TEPA provides best gene silencing efficiency [22]. Other researchers synthesize dendrimer-modified cyclodextrins and successfully deliver siRNA payload [23,24]. These data have motivated us to investigate the potential of TEPA-modified BCDs as carriers for DNA and siRNA.

In this study, we developed an easy and efficient strategy to generate TEPA-BCD. The 6-hydroxyl groups of BCD were activated with tosyl chloride. After nucleophilic displacement by ethylenediamine (EDA), TEPA-BCD nanoparticles were synthesized by the crosslinking reaction of EDA-BCD and TEPA via glutaraldehyde (GA). The synthesis of TEPA-BCD nanoparticles is demonstrated in Figure 1. We characterized fundamental parameters (zeta potential, morphology, size distribution, and molecular weight) using a dynamic light-scattering instrument and transmission electron microscopy. We revealed that these parameters play a role in the formation of polyplexes. TEPA-BCD enabled efficient transfection of plasmid DNA and siRNA with low cytotoxicity in a human retinal pigment epithelial cell line (ARPE) and a mouse embryonic fibroblast cell line (3T3). We demonstrated the ability of TEPA-modified BCD to form stable complexes with siRNA and provide efficient gene knockdown.

## 2. Results and Discussion

### 2.1. Functionalization and Characterization of TEPA-BCD Carrier

The FTIR spectra of BCD, EDA-BCD, and TEPA-BCD are shown in Figure 1A. All BCD derivatives (BCD, EDA-BCD, and TEPA-BCD) showed the characteristic bands of glucopyranoside: 2900 cm^−^^1^ (aliphatic C–H stretching vibration), 1036 cm^−^^1^ (C–O vibration). Another two intense peaks around 1036 and 2900 cm^−^^1^ from C-O and C–H stretching were also found in all BCD derivatives indicating the main BCD structure was maintained after the amine introduction [25]. The band at 3376–3423 cm^−1^ was assigned to the symmetric and asymmetric stretching vibrations of N–H and O–H bond stretching. The new peaks at 1020–1230 cm^−^^1^ accounted for C–N stretching in EDA- and TEPA-BCD but not in pure BCD, indicating the successful introduction of amine groups of EDA and TEPA on the BCD.

The high-resolution NMR spectrometer was also employed to characterize BCD and its derivatives. The ^1^H spectrum of Tosyl-BCD shows two signals at 7.75 and 7.45 ppm, indicating the benzene ring of tosyl group at glucose’s C-6. Our NMR results for Tosyl-BCD are close to those reported in previous publications [26,27]. The methyl proton on tosyl group had a signal at δ = 2.43 in ^1^H spectrum and δ = 31.2 in ^13^C spectrum (Appendix A). The CH_2_-NH-BCD at glucose’s C-6 had a new signal at δ = 2.1 in ^1^H spectrum and δ = 42.0 in ^13^C spectrum, indicating the replacement of tosyl group by EDA in EDA-BCD (Appendix A). Our EDA-BCD NMR spectra are close to those reported in Liu et al. [28]. The disappearance of benzene signal also confirmed this conclusion. The CH_2_-NH group in TEPA-BCD nanoparticles had the signals (^1^H δ = 2.84, ^13^C δ = 45.2) (Appendix A). The cyclic ether from the crosslinker of glutaraldehyde, mentioned in Migneault’s paper [29], gave the signals (^1^H δ = 2.22, ^13^C δ = 30.2) in Appendix A. The BCD structure is preserved after the conjugation since the existence of characteristic BCD signals (^1^H δ = 5.70, 4.84, 4.44, 3.48) and (^13^C δ = 101.8, 81.1, 73.0, 72.0, 60.2), which are in agreement with those reported in the paper by Jindřich’s group [27]. Our NMR data indicated the successful synthesis of TEPA-BCD nanoparticles.

We assessed the pH sensitivity of EDA-BCD and TEPA-BCD and their ability to undergo physicochemical amendments. In Figure 1B, we found that TEPA was an effective additive for increasing the zeta potential. Compared with EDA-BCD, TEPA-modified nanoparticles (TEPA-BCD) exhibited higher zeta potential from physiological pH to endo-lysosomal (acidic) environment. At pH 7, TEPA modification generated a positive zeta potential from about 3 to 16 mV. Surface zeta potential was affected upon a change in pH. TEPA-BCD surface charge was enhanced at pH 5 and pH 3. The results inferred the protonation of free primary amines of TEPA (containing five amines) and EDA (containing two amines) in acidic pH environment. TEPA could provide endo-lysosomal escape capability due to proton sponge effect and protection of loaded nucleic acids from the harsh endo-lysosomal environments [30,31]. The zeta potential data suggested successful coating of the TEPA around EDA-BCD nanoparticles. TEPA-BCD can bind nucleic acids with high affinity through electrostatic interactions.

We determined the morphology of BCD and their complexes using TEM (Figure 2). The new nucleic acid carrier can associate with plasmid DNA or siRNA and self-assemble into spherical nanoparticles. Our method produced EDA-BCD and TEPA-BCD in the range 149 to 243 nm in the dry state. We successfully prepared TEPA-BCD/plasmid DNA and TEPA-BCD/siRNA via electrostatic interaction. The size and zeta potential of TEPA-BCD complexes in the buffers are further investigated in the next section.

### 2.2. Effects of Complex Ratio on Physicochemical Properties of DNA- and siRNA-Loaded TEPA-BCD

The physicochemical properties of nucleic acid-loaded carriers, including particle size and surface zeta potential, can influence the efficacy of intracellular delivery and the gene expression manipulation [32,33]. These two properties can be tailored based on the ratio of anionic DNA/siRNA and cationic TEPA-BCD [34]. The polyplexes of nucleic acid–TEPA-BCD were denoted as TEPA-BCD/plasmid DNA and TEPA-BCD/siRNA. The particle size of these two polyplexes increased while their zeta potential decreased as the nucleic acid:nanoparticle ratio increased (Figure 3). The particle size of TEPA-BCD/plasmid DNA and TEPA-BCD/siRNA ranged from 795 to 332 nm (Figure 3A) and from 912 to 349 nm (Figure 3B) at high and low nucleic acid/carrier ratios. Loading of the negatively charged DNA and siRNA slightly decreased the zeta potential. Although it is not possible to exactly determine the position of the plasmid DNA and siRNA within these polyplexes, the decreased zeta potential data indicated NH_3_^+^ groups of TEPA-BCD were neutralized through electrostatic interaction by the presence of phosphate groups in nucleic acids. The increase in particle size and decrease in zeta potential after nucleic acid loading demonstrated the successful association of plasmid DNA and siRNA with the TEPA-BCD. The sizes of BCD complexes were large enough to avoid renal clearance, and also small enough to evade phagocytic uptake and clearance [35]. The positive zeta potential of TEPA-BCD/plasmid DNA and TEPA-BCD/siRNA nanoparticles may have resulted in the interaction with cell membrane, leading to improved drug delivery [36].

### 2.3. Transfection Efficiency and Viability of the TEPA-BCD Polyplexes

The efficiency of the TEPA-BCDs as DNA carriers was initially tested in ARPE cells and 3T3 cells by using a plasmid DNA containing gene-encoding green fluorescent protein (GFP) and anti-GFP siRNA (Figure 4). Polyplexes were prepared by fixing 1.28 μg (60 μL) of carrier and adjusting 0.001–1 μg of DNA and 1–40 ng of siRNA. The transfection efficiency was improved by increasing the ratio of plasmid DNA in loaded TEPA-BCDs. Only 0.075 μg of DNA was sufficient to reach 60% transfection efficiency in ARPE cells. The transfection efficiency of 90% was reached with polyplexes at TEPA-BCD/plasmid DNA ratio of 1.28:1. Moreover, higher transfection efficiency was observed for the TEPA-BCD/plasmid DNA in 3T3 cells. The transfection efficiency of 90% was reached with polyplexes at 1 μg of DNA. TEPA-BCD was able to transfect plasmid into these two cell lines and enabled plasmid-based gene expression. The gene-silencing efficiency of TEPA-BCD/siRNA nanoparticles was determined in APRE-GFP cells and 3T3-GFP (genetically engineered APRE cells and 3T3 cells expressing GFP), using an anti-GFP siRNA at 4 h post-transfection. While TEPA-BCD/siRNA showed gene-silencing effects in both cell lines, the silencing effect in 3T3-GFP cells was stronger than that in APRE-GFP cells (Figure 4B). The TEPA-BCD/siRNA reduced the level of GFP expression by 55%, at 10 ng of siRNA and 1.28 μg of TEPA-BCD. The cell viability of 3T3, ARPE, and their GFP expressing cells was above 75% (Appendix A) after the treatment of TEPA-BCD complexes.

### 2.4. Kinetics and Internalization Mechanisms of GFP Expression and Gene Silencing after DNA and siRNA Delivery

The transfection with GFP-encoded DNA in ARPE cells and 3T3 cells was carried out both with TEPA-BCD and a commercial reagent PolyJet (Figure 5). The fluorescence intensity was determined using the BioTek image system. We found that ARPE cell line was difficult to transfect with PolyJet. Treatment of 3T3 cells with DNA-loaded PolyJet did not result in good DNA delivery either. The expression of GFP was not observed in ARPE cell population after transfection and only started at 24 h post-transfection with a very slight increase in 3T3 cells. In contrast, the transfection with DNA-loaded TEPA-BCD, prepared at the DNA:nanoparticle ratio of 1:1.28 (*w*/*w*), showed a different relationship (ARPE cells in Appendix A and 3T3 cells in Appendix A). The expression of GFP started before 4 h with a steep increase, reaching 100% at 24 h (Figure 5), suggesting that cells had taken up the nanoparticles quickly. In ARPE cells, the GFP expression was transient. In 3T3 cells, however, the GFP expression was steadily maintained for 2 days, and the fluorescence intensity was 10 times higher than the initial value. The optimized cationic agent TEPA-BCD could be used to improve cellular uptake and was a good carrier of plasmid DNA for gene expression [13]. For gene silencing, we compared the transfection efficiency of ARPE-GFP cells and 3T3-GFP cells both with TEPA-BCD and the commercial transfection reagent GenMute. The gene-silencing efficiency was determined by the same method: comparing fluorescent intensity and the number of GFP expressing cells. The control was represented by untreated cells. It must be noted that the control untreated cells were continuously dividing (Appendix A), resulting in increased fluorescence in the control (Figure 6). We found that the cell proliferation was not constrained by the presence of TEPA-BCD nanoparticle or GenMute. For the GenMute experiment, there was little reduction in GFP expression observed in ARPE-GFP cells. The silencing of GFP in 3T3-GFP cells was observed at 24 h, but the silencing effect was found to be transient. In contrast, TEPA-BCD/siRNA showed considerably stronger gene-silencing effects. Gene silencing started immediately after treatment, and silencing efficiency was continuously increased. After 4 h, the gene-silencing efficiencies were ~40% (in ARPE-GFP cells) and ~44% (in 3T3-GFP cells), respectively. The results validated the use of TEPA-BCD delivery for improving siRNA activity. The cationic carriers may traffic into cells through lysosomes [37]. We assessed the colocalization between FITC-labeled TEPA-BCD and lysosomes. In both ARPE cell and 3T3 cells, we observed that FITC probes (green) were colocalized with the LysoTracker Red (red), which accumulated in late endocytic structures (Figure 7A,B). The images showed a mixture of TEPA-BCD in lysosomes (yellow) and a small population of lysosomes without TEPA-BCD (red). Some TEPA-BCDs (green) existed in the cytoplasm and did not reach lysosomes. Finally, the effects of five endocytic inhibitors on the complexes of TEPA-BCD and nucleic acids are summarized in Table 1 and Figure 7C,D. We found that these results are consistent with a model in which TEPA-BCD uptake proceeds through an endocytic pathway that is energy dependent and requires clathrin (for both plasmid DNA and siRNA delivery) and caveolin (for siRNA delivery only).

## 3. Discussion

Several parameters such as charge density, hydrophilic–hydrophobic balance, nature of the functional groups, and spacer length of BCD have been reported to affect the binding capacity of nucleic acids and cell transfection [38]. There has been growing interest in the development of BCD as delivery vehicles for siRNA. For example, an amphiphilic BCD knockdowns 80% of macropinocytosis gene (PAK1) expression using the BCD/siRNA (1 µM/100 nM) in glioblastoma cells [39]. Twenty percent of NF-κB transcription factor is inhibited using cationic cyclodextrin and anti-RelA siRNA in prostate cancer cells [40]. About 43% of PLK1 expression is knockdown by 50 nM of cationic BCD-siRNA conjugate using qRT-PCR to quantify in prostate cancer cells [41]. Hyaluronate-modified α-Cyclodextrin nanoparticles inhibit GAPDH gene with a silencing efficiency of 55% using 1000 nM of siRNA in human lung adenocarcinoma cells [42]. Folate-cyclodextrin-modified dendrimer can deliver siRNA and inhibit 40% of luciferase activity in KB cells transiently expressing luciferase [43].

We first hypothesized that introduction of cationic moieties and crosslinking of BCDs may enhance their delivery capacity for both DNA and siRNA. Our results indicated TEPA-BCD had higher positive charge at the physiological pH, which resulted from the primary amines of TEPA and EDA. These cationic groups can electrostatically interact with the phosphate groups in nucleic acids to form the nanoparticles. The morphology and sizes of BCD nanoparticles demonstrated that the carrier adsorbed DNA and siRNA in a spherical shape. The payload of the DNA and siRNA decreased the zeta potential slightly. The loading ratio of nucleic acids on TEPA-BCD could impact the efficacy of intracellular delivery, silencing, and transfection efficiency in all tested cells in this study.

Secondly, we hypothesized that the BCDs developed could affect endocytosis pathways through the interaction with membrane lipids and receptors. To understand the endocytosis during the cell entry of TEPA-BCD carriers, we examined the effects of five endocytic inhibitors: inhibitors of clathrin-mediated endocytosis (chlorpromazine (CPZ) and monensin (Mon)); an inhibitor of caveolae-mediated endocytosis (genistein (Gen)); a lipid rafts/caveolae-dependent endocytosis inhibitor (methyl-β-cyclodextrin (MCD)); and an ATPase inhibitor (sodium azide (SA)) (Table 1) [44,45]. In TEPA-BCD/plasmid DNA in vitro internalization process, caveolae-mediated endocytosis was shown to be not involved, since Gen and MCD appeared to have no inhibitory effect (Figure 7C). However, the transfection efficiencies were reduced in the preincubation with CPZ, Mon, and SA in APRE cells and 3T3 cells, compared with the nontreatment controls (Figure 7D).

CPZ blocked clathrin-coated pit formation [46], and considerably decreased the TEPA-BCD/plasmid DNA uptake. As an ionophore, Mon can neutralize the pH gradient of endocytic vesicles. Mon inhibitor decreased the polyplex uptake, indicating that endosomes were involved in the intracellular transport of the polyplexes. The highest degree of inhibition was observed for both CPZ and Mon in APRE cells (>60%). The contrasting effect observed with CPZ/Mon and Gen/MCD suggested that clathrin-mediated endocytosis was the predominant uptake pathway of TEPA-BCD/plasmid DNA. We also found that the uptakes of both TEPA-BCD/plasmid DNA (Figure 7C) and TEPA-BCD/siRNA (Figure 7D) were energy dependent, because in the presence of SA, these two cell lines took up DNA and siRNA less efficiently.

Development of siRNA delivery systems depends not only on their transfection efficiencies, but also on their gene-silencing effect. Although both Gen and MCD treatments showed no reduction in plasmid DNA transfection efficiency (Figure 7C), the gene-silencing effect of TEPA-BCD/siRNA was affected by the presence of these two inhibitors (Figure 7D). The 100 times lower molecular weight of siRNA, compared to plasmid DNA, may present fewer obstacles for delivery and thus siRNA carriers can undergo internalization via caveolae-mediated endocytosis. To examine the clathrin-mediated route of cell entry of TEPA-BCD/siRNA, we made use of CPZ and Mon agents. Although Mon inhibited plasmid DNA delivery (Figure 7C), its presence did not affect the gene silencing of siRNA (Figure 7D). Mon inhibited acidification of lysosomes, but it did not affect the distribution of clatherin-coated pits. We found that gene silencing was sensitive to CPZ, suggesting clatherin-coated pits formation was important in the process. Since the cells derived from different origins could have distinct endocytosis pathway and knockdown efficiency, TEPA-BCD can be successfully applied to two kinds of cells. A coculture model using prostate cancer PC3 cells and osteoblast hFOB cells to simulate prostate cancer metastasis to bone has been used to evaluate the anti-RelA siRNA delivery by cationic BCD [40]. In that study, the host of siRNA delivery is PC3 cells but not bone cells. The knockdown efficiency of NF-κB in PC3 cells is 50% using 2D coculture and 25% in 3D system. In this study, the TEPA-BCD developed can introduce both siRNA and plasmid into cells and successfully express in two cells.

In the intravenous delivery, the nanoparticles have to escape the various barriers such as kidney filtration, phagocytosis, and hydrolysis degradation. Sequentially, they should transport across vascular vessels, diffuse through the extracellular matrix, and reach the target cells [47]. The enhanced permeability and retention effect contributes to the size-dependent accumulation of nanoparticles in the tumor. Some solid tumors have leaky vascular tubes permitting the accumulation of hundred-nanometer nanoparticles in tumors. Highly permeable tumors, such as the colon adenocarcinoma LS174T, are reported to allow significant accumulation of 400 nm nanoparticles [48]. The size of nanocarriers should be maintained during circulation until the tumor region is reached. Nonionic and hydrophilic brushes such as poly ethylene glycol and poly vinyl pyrrolidone can generate steric repulsive forces to repel the adsorption of serum proteins and proteoglycans on the nanoparticle. Targeted delivery is another issue in therapeutic siRNA delivery. The transferrin receptor is expressed on some cancers and the transferrin has been chosen as a ligand for tumor-targeted delivery. Cyclodextrin-based cationic polymer (CALAA-01) in clinical trials has been modified with transferrin and PEG to target the melanoma and prevent the aggregation in the circulation system [49]. Based on the aforementioned design, the target ligand and polymer brushes should be incorporated in the nanoparticles of TEPA-BCD to escape the physiological barriers in our future animal tests.

In summary, we developed a cationic BCD-based nanoparticle delivery system and applied it in the fibroblast 3T3 cell line and the epithelial ARPE cell line. Our results demonstrated the multifunctionality critical for internalization and activity of both gene-coded plasmid DNA and siRNA. When the TEPA-BCD/plasmid ratio was 1.71:1, the nanoparticles had a transfection efficiency of 97%, and cells maintained viability around 83%. When the TEPA-BCD/siRNA ratio was 96:1, it knocked down the fluorescent expression by 57%, yet the cell viability remained around 82%. TEPA-modified BCD demonstrated a nanometer size, positive surface charge, minimal cytotoxicity, great electrostatic interaction with nucleic acid, and efficient delivery. In both ARPE cells and 3T3 cells, TEPA-BCD entered cells through clathrin- and caveolae-mediated endocytosis. TEPA-BCD/plasmid DNA mediated high gene expression and TEPA-BCD/siRNA displayed efficient gene silencing. The TEPA-BCD provides a promising platform for delivery of therapeutic nucleic acids in vitro and warrants further development of its potential for in vivo delivery.

## 4. Materials and Methods

### 4.1. Materials

BCD was a generous gift from Feng-Yuan Biotech (Jiangsu, China). Acetone, methanol, and ethanol were purchased from Echo Chemicals (Taoyan, Taiwan). Green fluorescent protein (GFP)-expressing plasmid was obtained from Takara Bio (GFP-C3, Shiga, Japan). Fetal bovine serum (FBS) was purchased from Biological Industries (Haemek, Israel). The siRNA against GFP gene (sense: 5′-GCAGCACGACUUCUUCAAGdTdT-3′; antisense: 5′-CUUGAAGAAGUCGUGCUGCdTdT-3′) was purchased from MDBio (Taipei, Taiwan). DNA and siRNA transfection controls (PolyJet and GenMute) are purchased from SignaGen (Rockville, MD, USA). TEPA, p-tolylsulfonyl chloride, sodium chloride, and other chemicals were purchased from Sigma-Aldrich (St. Louis, MO, USA). All reagents were used without further purification.

### 4.2. TEPA-BCD Synthesis

Mono-6-deoxy-6-p-tolylsulfonyl (tosyl)-BCD and EDA-BCD were synthesized as described previously [28,50]. Briefly, 6 g BCD was suspended in 50 mL of double-distilled water, and 2 mL of 16.4 N NaOH was added drop by drop to the suspension within 6 min. The suspension became slightly yellow and homogenous, after which 1.08 g of tosyl chloride in 3 mL of acetonitrile was added to modify on the 6-hydroxyl group of BCD for 3 h. Three cycles of a low-temperature precipitation and a high-speed centrifugation were used to collect the product. The recovered Tosyl-BCD (yield = 8.5%) was reacted with EDA in a 1:1 molar ratio to form EDA-BCD. Cold acetone was added to the mixture to precipitate the white EDA-BCD (yield = 79%). The mixture of 10 mg EDA-BCD, 23.6 mg TEPA, and 50 μL GA (1.3%) was then vortexed for 2 h at room temperature to obtain TEPA-BCD nanoparticles (yield = 16.5%). The remaining aldehyde group was quenched by the addition of 0.1 mg/mL glycine and vortexed for 30 min. The unreacted reagents were removed by washing with deionized water twice using the 50 kDa cutoff Vivaspin column (GE, Marlborough, MA, USA) for purification. The NMR chemical shifts of BCD, Tosyl-BCD, EDA-BCD, and TEPA-BCD are summarized as follows. ^1^H NMR (600 MHz, DMSO-d6) of BCD: δ = 5.69 (OH-2, OH-3, Integral = 2.00), 4.84 (H-1, Integral = 1.01), 4.44 (H-6, Integral = 1.00), 3.62 (H-2, H-3, H-4, Integral = 4.11), 3.33 (H-5, H-6, Integral = 6.13) ppm. ^13^C NMR (600 MHz, DMSO-d6) of BCD: δ = 102.04 (C-1), 82.01 (C-4), 73.51–72.51 (C-2, C-3, C-5), 69.63 (C-6I, C-5I), 60.37 (C-6) ppm. ^1^H NMR (600 MHz, DMSO-d6) of Tosyl-BCD: δ = 7.12 (aromatic H-2′, Integral = 1.95), 7.45 (aromatic H-3′, Integral = 2.74), 5.70 (OH-2, OH-3, Integral = 13.92), 4.80 (H-1, Integral = 25.84), 4.50 (OH-6, Integral = 16.40), 4.32 (H-6aI, Integral = 3.96), 4.21 (H-6bI, Integral = 3.18), 3.48 (H-2, H-3, H-4, H-5, H-6, Integral = 210.35), 2.43 (H-5′, Integral = 1.29) ppm. ^13^C NMR (600 MHz, DMSO-d6) of Tosyl-BCD: δ = 128.48 (C-3′, C-2′), 102.44 (C-1), 82.02 (C-4), 73.55–72.50 (C-2, C-3, C-5), 60.41 (C-6), 31.16 (Tosyl-CH_3_) ppm. ^1^H NMR (600 MHz, DMSO-d6) of EDA-BCD: δ = 5.70 (OH-2, OH-3, Integral = 76.74), 4.84 (H-1, Integral = 40.12), 4.44 (H-6, Integral = 33.34), 3.48 (H-2, H-3, H-4, H-5, H-6, Integral = 472.52), 2.1 (–CH_2_NH–BCD, Integral = 25.10) ppm. ^13^C NMR (600 MHz, D2O) of EDA-BCD: δ = 101.82 (C-1), 81.09 (C-4), 73.04–71.78 (C-2, C-3, C-5), 60.25 (C-6), 42.04 (BCD-C-C-N) ppm. ^1^H NMR (600 MHz, D2O) of TEPA-BCD: 5.06 (OH-2, OH-3, Integral = 18.29), 4.77 (H-1, Integral = 5.67), 4.47 (H-6, Integral = 1.00), 3.74 (H-2, H-3, H-4, Integral = 126.44), 3.19 (H-5, H-6, Integral = 59.43), 2.84 (CH_2_-NH, Integral = 36.51), 2.22 (cyclic ether C-C-O, Integral = 59.41) ppm. ^13^C NMR (600 MHz, D2O) of TEPA-BCD: δ = 102.10 (C-1), 81.24 (C-4), 73.20–72.08 (C-2, C-3, C-5), 60.15 (C-6), 45.22 (C-NH), 30.24 (cyclic ether C-C-O) ppm.

### 4.3. Preparation of TEPA-BCD/Plasmid DNA and TEPA-BCD/siRNA

We transformed plasmid GFP-C3 into DH-5α *E. coli* cells. The host *E. coli* was grown in LB broth with kanamycin (50 µg/mL) and the plasmid DNA was isolated using Mini Purification Kit (Genemark, Taipei, Taiwan). The concentration and quality of nucleic acids were determined using a NanoDrop spectrophotometer (Thermo, Framingham, MA USA) and the Beer–Lambert law with an extinction coefficient of 50 μg/mL^−1^ cm^−1^ at 260 nm. Isolated plasmid DNA had OD 260/280 ratios of 1.80–1.95, indicating the quality was suitable for our application. Polyplexes were formulated by mixing 0.21–1.71 μg (10–80 μL) TEPA-BCD with plasmid DNA (1 μg) or siRNA (13.3 ng, 10 nM) in 100 μL DMEM medium.

### 4.4. Characterization of TEPA-BCD

To visualize morphology of BCD derivatives, diluted samples were adsorbed onto a 100 mesh copper grid and dried overnight, followed by imaging on a transmission electron microscope (TEM, JM-1011, JEOL, Tokyo, Japan). The magnification power of TEM is 50,000. The chemical groups of modified BCDs were investigated using Fourier transform infrared spectroscopy (FTIR, Alpha, Bruker, Berlin, Germany) and an NMR spectrometer (Bruker AV III HD). The size and zeta potentials of the TEPA-BCD complexes were measured using ZetaSizer with 633 nm He-Ne Laser (Nano ZS 90, Malvern, MA, USA) and the folded capillary cell (Malvern part no. DST1070).

### 4.5. GFP Silencing and Transfection In Vitro

ARPE cells (a human retinal pigment epithelial cell line) and 3T3 cells (a mouse embryonic fibroblast cell line) from the cell bank of BCRC (Hsinchu, Taiwan) were used for GFP plasmid delivery experiments. DNA transfection was performed as previously reported [22]. APRE-GFP cells and 3T3-GFP (genetically engineered APRE cells and 3T3 cells that express GFP) from the National RNAi Core Facility at Academia Sinica (Taipei, Taiwan) were used for siRNA delivery experiments. ARPE and 3T3 cells were grown in DMEM/F12 and DMEM medium (high glucose) supplemented with 10% FBS, respectively. Cells were seeded in 96-well culture plates at 20,000 cells/well. When reaching 80–90% confluence, cells were transfected with 100 µL of freshly prepared polyplexes in serum-free DMEM medium and incubated for 6 h at 37 °C, the polyplex aspirated, and FBS-containing medium added. Cells were cultured for 24 h. The cells were washed with PBS and fixed with ethanol/acetic acid solution (5% acetic acid, 95% ethanol) for 5 min at room temperature. The fixed cells were then stained with 100 µL of Hoechst 33,342 (10 ppm) for 20 min. Fluorescence intensity and cell viability were observed with an automated image analyzer (Lionheart FX, Biotek, VT, USA) using the excitation/emission (480/525 nm) and 10× magnification. To monitor GFP expression kinetics, cells were transfected and monitored up to 48 h at 37 °C using the BioTek system implemented with temperature and humidity control. The transfection and knockdown efficiencies were quantified by using the BioTek image analyzer. Two wells were analyzed and four images per well were captured in every experiment. Finally, the results from two individual experiments were averaged and the standard deviation calculated. For each experiment, a total of eight images (100–400 cells per image) from two wells were acquired to reduce the variance.

### 4.6. Effects of Endocytic Inhibitors on Transfection and Gene-Silencing Efficiency

The cellular uptake mechanisms of polyplex were examined using different endocytic inhibitors: chlorpromazine (30 µM), genistein (200 µM), sodium azide (10 mg/mL), methyl-BCD (10 mM), and monensin (3 µM). The cells were seeded in 48-well plates at 5 × 10^4^ cells/well. When reaching 90% confluence, the cells were treated with polyplex and endocytic inhibitor for 3 h. Polyplexes were formulated by mixing 1.28 μg TEPA-BCD with plasmid DNA (1 μg) or siRNA (13.3 ng) in 100 μL DMEM medium. Subsequently, the transfection medium was replaced with fresh DMEM supplemented with 10% FBS and cells were incubated for another 24 h. Imaging was conducted on an automated image analyzer for analysis of fluorescence intensity and cell viability.

### 4.7. Statistical Analysis

All cellular data were performed by two individual experiments. The Zetasizer results were the average from three individual experiments. The mean and standard deviation were calculated using Microsoft Excel. Statistical comparisons are performed using PROC ANOVA followed by the Tukey post hoc test using SAS 9.4 software (Cary, NC, USA) and unpaired two-tailed Student *t*-test by Analysis Toolpak in Microsoft Excel. Statistical significance is represented as * *p* < 0.05.

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
