# Peer review of "Tetraethylenepentamine-Coated β Cyclodextrin Nanoparticles for Dual DNA and siRNA Delivery"

_pharmaceutics, 2022, doi:10.3390/pharmaceutics14050921_

Round 1

Reviewer 1 Report

  1. The introduction part could be written in a better form. Please improve it.
  2. Please add the brand and manufacturer country of TEM.
  3. Which software was used for Student's t-test?
  4. Please rewrite the FTIR result section. You should mention the differences between each material after each functionalization step.
  5. "We determined morphology and particle sizes of BCD nanoparticles using TEM and Zetasizer (Fig 2)." Please delete the Zeta sizer in here.
  6. Please redraw the curves of figure 3 based on the nucleic acid: nanoparticle ratio.
  7. There are some grammatical mistakes in the text which should be corrected.
  8. Based on your results "lysosomes were involved in the intracellular transport of the polyplexes" so how did they have no effect on the loaded DNA and siRNA?
  9. The discussion part of the manuscript should be improved. You just add your results there without any comparison with other similar works.
  10. Please transfer what you mentioned in lines 268-283 to the first part of the results.
  11. "TEPA-modified BCD demonstrated a nanometer size, positive surface charge, minimal cytotoxicity," You didn't test the cytotoxicity effect of nanoformulations.     

Reviewer 2 Report

Dear authors,

This manuscript aims to gain insights into dual targeting strategies using nano systems and puts forward a well-explained research project for the same.

In lieu of the current manuscript, please find below some review comments and suggestions:

What was the purpose of the study for specifically utilizing only 3T3 and ARPE cells where one is a fibroblast cell line and the other being and epithelial cell line? Is there any specific target that influenced the use of these 2 specific cell lines? At the same time, would this dual target formulation be of application to a broader range of cell lines – for example in various other disease conditions including various cancers? This is of utmost importance to define (and perform experiments using wider range of cell lines if feasible) as different diseases put forward a wide range of heterogeneity that could limit the use of particular nano systems. It would also enable the readers to apply this knowledge while designing similar nanoformulations for other targets at a broader prospect.

Are there any examples of such dual systems currently in clinical practice or pipeline? It would be of worth to put forward a brief summary on the current status of such formulations in the introduction or at the beginning of the discussion.

Other minor checks:

Line 49: check and correct spelling of membrane

Line 53: Check and correct spelling of nanocarriers

Line 61: Check and correct spelling “protecting”

Please edit all “in vivo” and “in vitro” to italic font.

Please consider proof-reading for any grammatical and typo errors.

Reviewer 3 Report

The article presents the data on cationic cyclodextrin nanoparticles as  the efficient carriers for both pDNA and siRNA. They synthesized nanopacticles and checked the efficiency of them to trasfer pDNA and siRNA into cells. The article is well written and easy to read and understand. The data are mainly  correctly done and well described. The results are well discussed and will be interesting for wide audience. As for English, I am not native speaker and for me it was acceptable.

I have only minor remarks:

1) Please, add the conclusion section after the discussion section.

2) In the discussion section the part starting from "The high-resolution NMR spectrometer (Bruker AV III HD) ..." belong to Results section or Mat&Meth.

3) The statistical analysis of Fig. 6 should be made by ANOVA and multiple comparison tests (but not two-samples Student's test).

4) Fig. 4: n=2 is too small for any statistical analysis. The data are not statistically significant. 

Reviewer 4 Report

The authors of the study synthesized novel tetraethylenepentamine-coated β-cyclodextrin (TEPA-BCD) nanoparticles which could be used to deliver nucleic acids (plasmid or siRNA species) for in vitro and in vivo  applications (e.g., gene transfer, gene silencing, etc.). The TEPA-BCD nanoparticles were further characterized (i) for their loading efficiency of nucleic acids, (ii) structurally for their zeta potential, size and morphology, and (iii) functionally for their ability to deliver nucleic acids into cells in vitro. Regarding the latter, the authors further compared the ability of TEPA-BCD nanoparticles to deliver nucleic acids to a few commercially available polycationic reagents. Lastly, the authors mechanistically connected the internalization of TEPA-BCD nanoparticles to the clathrin- and caveolin-dependent endocytosis pathways in studies where they used chemical inhibitors of these pathways. While the authors provide convincing data on the structural characterization of their TEPA-BCD nanoparticles, I have several comments and recommendations as following.    

  • The in vivo translatability of these nanoparticles is expected to be low. One main reason for this is the average size of their nucleic acid-loaded TEPA-BCD nanoparticles (for both plasmid and siRNA species), which is fairly large (i.e., larger than 200 nm in diameter). The average size of these nanoparticles makes their potential usage for in vivo applications (such as the delivery of nucleic acids to tumors, etc.) rather questionable. This is because it is unclear how nanoparticles of this size will be able to penetrate the tumor vasculature efficiently for passive delivery of nucleic acids into the tumor beds. This is an important limitation of the present study’s findings, which should be addressed in the ‘Discussion’ section along with potential mitigating strategies that might be envisioned by the authors to alleviate some of these translation issues.
  • No data was provided by the authors to show whether their TEPA-BCD nanoparticles have any impact on cell viability. Most polycationic transfection reagents (e.g., polybrene, etc.) are demonstrably cytotoxic at higher concentrations and, for this reason, it would be important to see how the TEPA-BCD nanoparticles compare to some of the popular commercially available transfection regents in side-by-side comparisons. Moreover, the transfection efficiency of TEPA-BCD nanoparticle was not compared to another gold standard method (i.e., electroporation).  
  • For the fluorescent microscopy studies that were done to visualize the uptake of FITC-labeled TEPA-BCD nanoparticles and their co-localization with lysosomes, it is recommended to capture much higher magnification images in order to visualize these intracellular co-localization events more clearly. The images provided by the authors only tell a very incomplete story in this regard.
  • Finally, the chemicals used to block the endocytosis pathways in the mechanistic studies that were done by the authors are only blocking these pathways in a non-specific manner and require extremely high concentrations (i.e., in the mM range) for these effects to take place. This is an important methodological issue because the high concentrations at which these chemicals are used could (and are expected to) impact a number of additional intracellular processes in additional to endocytosis. My recommendation for the authors is to use in their internalization studies more specific inhibitors, such as the commercially available dynamin inhibitors (e.g., the non-competitive Dynasore or selective MiTMAB and Dyngo-4a inhibitors) which are known to inhibit endocytosis with higher specificity and potency (i.e., in the low μM range).                

Reviewer 5 Report

Ms. Ref. No.: pharmaceutics-1644762
Title: Tetraethylenepentamine-coated β cyclodextrin nanoparticles for 2 dual DNA and siRNA delivery

The manuscript describes the use of tetraethylene-pentamine-coated β-cyclodextrin (TEPA-BCD) nanoparticle to be evaluated for the targeted cell delivery of plasmid DNA and siRNA. Characterization of the TEPA-BCD is well done and authors concluded that TEPA-BCD/plasmid DNA mediated high gene expression and TEPA-BCD/siRNA displayed efficient gene silencing. Although data are interesting and well obtained, there are some problems with this paper that should be considered:

The authors have made a very brief writing that affects many parts of the manuscript. To begin with, the introduction is very brief. But there is a large experimental part poorly described and in the absence of accessories. For example, line 333 mentions the concentration of nucleic acids but does not indicate how it is determined. In the TEM measurements the type of equipment used or the power is not specified, in the zeta potential measurements the cuvettes used are not specified nor the type of laser. On the other hand, line 121 indicates that the sizes are 169 with an error of 13 and 194 with an error of 8 nm. But how are these errors determined? There is no histogram and no count number high enough to apply an error calculation with any accuracy is indicated. Finally, although as minor errors, the authors should improve the presentation of the figures. There is no uniformity in sizes (see supplementary material) and there are graphics that are not centered in the text. On the other hand, there are references that can be replaced by more modern ones (see, for example, ref. 26 and 30). Ultimately, the authors should improve the introduction and better clarify the originality of the manuscript. The writing and presentation is also improvable and the experimental part is very expandable. A clarification is necessary to know with certainty the determination of the size of the clusters and the origin of the error in its 

Round 2

Reviewer 1 Report

-

Reviewer 2 Report

The authors have made all the mentioned changes.

Reviewer 4 Report

I thank the authors for their efforts to address my recommendations and make further improvements to their manuscript. I think the manuscript reads much better now, although there is still room for improvement. Nonetheless, I recommend the manuscript to be accepted for publication in its present form.  

Reviewer 5 Report

The manuscript has been corrected and can be published.

This manuscript is a resubmission of an earlier submission. The following is a list of the peer review reports and author responses from that submission.

Round 1

Reviewer 1 Report

In the resubmitted manuscript, the authors have made significant improvement, especially in characterization of the structures. 1H NMR and 13C NMR are provided and the results can be well supported. Thus I'd like to recommend the acceptance of this revised manuscript.

Author Response

Many thanks to the valuable suggestions.

Reviewer 2 Report

Dear authors,

I really appreciate your effort to better describe NMR spectra. However, my opinion is still the same. Your compounds seem not to be the ones you present.

Let me explain:

In case of Tosyl-b-CD 1H NMR, there are two other aromatic (tosyl) peaks (approx. 7.1 and 7.5 ppm, which are not integrated). Therefore, there must be at least two different tosyl groups. Probably one reacted and one unreacted. Moreover, the integrals are wrong (same as in the other 1H spectra).

There must be also more carbon atoms visible in 13C spectra. Please compare e. g. with the previously suggested literature (see Supporting information there):

Popr, M.; Hybelbauerova, S.; Jindrich, J. A Complete Series of 6-Deoxy-Monosubstituted Tetraalkylammonium Derivatives of Alpha-, Beta-, and Gamma-Cyclodextrin with 1, 2, and 3 Permanent Positive Charges. Beilstein J. Org. Chem. 2014, 10, 1390–1396. https://doi.org/10.3762/bjoc.10.142.

In case of EDA-CD, the 13C NMR spectrum is very similar to native b-CD, suggesting it is only native b-CD (together with dissolved ethylenediamine). Again, there should be more carbon atoms in 13C spectra. Please compare with similar compounds described in the abovementioned article.

Methods like HPLC-MS and 2D NMR spectra (especially HMBC), should unambiguously confirm the structures of your substances.

Author Response

Comment 1 of the reviewer:

In case of Tosyl-b-CD 1H NMR, there are two other aromatic (tosyl) peaks (approx. 7.1 and 7.5 ppm, which are not integrated). Therefore, there must be at least two different tosyl groups. Probably one reacted and one unreacted. Moreover, the integrals are wrong (same as in the other 1H spectra). There must be also more carbon atoms visible in 13C spectra. Please compare e. g. with the previously suggested literature (see Supporting information there): Popr, M.; Hybelbauerova, S.; Jindrich, J. A Complete Series of 6-Deoxy-Monosubstituted Tetraalkylammonium Derivatives of Alpha-, Beta-, and Gamma-Cyclodextrin with 1, 2, and 3 Permanent Positive Charges. Beilstein J. Org. Chem. 201410, 1390–1396.

Author response:

We appreciate the valuable suggestion. The integral ratio of 1H chemical shift at δ=7.45 (Integral=2.74) versus δ=7.12 (Integral=1.95) of Tosyl-BCD is close to 1:1 indicating the Tosyl-BCD has been synthesized. Two signals at 7.45 and 7.12 ppm indicating the two kinds of hydrogen on tosyl group at glucose’s C-6. Our 1H NMR results (Fig R1) for Tosyl-BCD are similar to the paper of JindÅ™ich’s group (Fig R2) [1]. However, the pattern of chemical shifts from JindÅ™ich’s paper was not exactly the same as the Petter’s original paper (1H NMR of Tosyl-BCD: δ 7.74(= 8.07 Hz, 2 H), 7.42 (d, J = 8.02 Hz, 2 H), 5.87-5.58 (m, 14 H), 4.82 (br s, 4 H), 4.76 (br s, 3 H), 4.55-4.13 (m, 6 H), 3.74-3.43 (m, 28 H), 3.42-3.18 (m, overlaps with HOD), 2.42 (s, 3 H) ppm) [2]. There are two unclear peaks around 1.2 and 2.2 ppm. Our NMR signal of 13C spectrum (Fig R3) was not clear compared to the data of JindÅ™ich’s group (Fig R4). In addition, the 13C chemical shifts of Tosyl-BCD from JindÅ™ich’s paper (Fig R4) was similar to the Petter’s paper (13C NMR: δ144.9 (s), 132.8 (s), 129.8 (d), 127.8 (d), 102.1 (m), 81.8 (d), 73.3-71.4 (m), 70.0, 68.7, 59.5 (t), 21.1 (4) ppm). However, two minor peaks around 125 and 29 ppm in Fig R4 [1] are not mentioned by Petter’s paper. These noises may be due the chemicals, the sample treatment and machine condition.

Comment 2 of the reviewer:

In case of EDA-CD, the 13C NMR spectrum is very similar to native b-CD, suggesting it is only native b-CD (together with dissolved ethylenediamine). Again, there should be more carbon atoms in 13C spectra. Please compare with similar compounds described in the abovementioned article. Methods like HPLC-MS and 2D NMR spectra (especially HMBC), should unambiguously confirm the structures of your substances.

Author response:

We appreciate the valuable suggestion. We carefully solubilize and precipitate the EDA-BCD product several times to reduce the EDA contamination. Compared to the BCD (Fig R5), 13C NMR of EDA-BCD (Fig R6) shall have 2 carbon signals from EDA (N-C1-C2-N). Since C1 and C2 are really close, we only have a new signal at δ=42.07. Similarly, the 1H NMR of EDA–BCD at BCD-N-CH2-CH2-N had a new signal at δ=2.1 as compared to BCD (Fig R7). Our EDA-BCD 1H NMR pattern (Fig R8) is close to the paper of Liu et al, who’s 1H NMR containing δ 4.90 (7H, C(1)–H), 3.83–3.68 (28H;C(3)–H, C(6)–H, C(5)–H), 3.51–3.25 (14H; C(2)–H, C(4)–H), 2.89 (2H, –CH2NH–BCD) [3]. Finally, we want to emphasize the NMR spectra are affected by several factors including the NMR condition, original materials, and operator expertise. In some cases, the same compound has slight difference in the published spectra. This manuscript provides the spectra of BCD, Tosyl-BCD, EDA-BCD, and TEPA-BCD that can compare with each other to prove the successful synthesis.  The synthesized TEPA-BCD nanoparticles also have been successfully applied as the DNA and siRNA nanocarrier.
